# Gaussian Prototypical Networks for Few-Shot Learning on Omniglot

## Abstract

We propose a novel architecture for $k$-shot classification on the Omniglot dataset. Building on prototypical networks, we extend their architecture to what we call *Gaussian prototypical networks*. Prototypical networks learn a map between images and embedding vectors, and use their clustering for classification. In our model, a part of the encoder output is interpreted as a confidence region estimate about the embedding point, and expressed as a Gaussian covariance matrix. Our network then constructs a direction and class dependent distance metric on the embedding space, using uncertainties of individual data points as weights. We show that Gaussian prototypical networks are a preferred architecture over vanilla prototypical networks with an equivalent number of parameters. We report results consistent with state-of-the-art performance in 1-shot and 5-shot classification both in 5-way and 20-way regime on the Omniglot dataset. We explore artificially down-sampling a fraction of images in the training set, which improves our performance. Our experiments therefore lead us to hypothesize that Gaussian prototypical networks might perform better in less homogeneous, noisier datasets, which are commonplace in real world applications.

## 1 Introduction

### 1.1 Few-shot learning

Humans are able to learn to recognize new object categories on a single or small number of examples. This has been demonstrated in a wide range of activities from hand-written character recognition (Lake et al., 2011), and motor control (Braun et al., 2009), to acquisition of high level concepts (Lake et al., 2015). Replicating this kind of behavior in machines is the motivation for studying few-shot learning.

Parametric deep learning has been performing well in settings with abundance of data. In general, deep learning models have a very high functional expressivity and capacity, and rely on being slowly, iteratively trained in a supervised regime. An influence of a particular example within the training set is therefore small, as the training is designed to capture the general structure of the dataset. This prevents rapid introduction of new classes after training. (Santoro et al., 2016)

In contrast, few-shot learning requires very fast adaptation to new data. In particular, $k$-shot classification refers to a regime where classes unseen during training must be learned using $k$ labeled examples. Non-parametric models, such as $k$-nearest neighbors (kNN) do not overfit, however, their performance strongly depends on the choice of distance metric. (Atkeson et al., 1997) Architectures combining parametric and non-parametric models, as well as matching training and test conditions, have been performing well on $k$-shot classification recently.

### 1.2 Gaussian prototypical networks

In this paper we develop a novel architecture based on prototypical networks used in Snell et al. (2017), and train it and test it on the Omniglot dataset (Lake et al., 2015). Vanilla prototypical networks map images into embedding vectors, and use their clustering for classification. They divide a batch into *support*, and *query* images, and use the embedding vectors of the support set to

define a class prototype – a typical embedding vector for a given class. Proximity to these is then used for classification of query images.

Our model, which we call the *Gaussian prototypical network*, maps an image into an embedding vector, and an estimate of the image quality. Together with the embedding vector, a confidence region around it is predicted, characterized by a Gaussian covariance matrix. Gaussian prototypical networks learn to construct a direction and class dependent distance metric on the embedding space. We show that our model is a preferred way of using additional trainable parameters compared to increasing dimensionality of vanilla prototypical networks.

Our goal is to show that by allowing our model to express its confidence in individual data points, we reach better results. We also experiment with intentionally corrupting parts of our dataset in order to explore the extendability of our method to noisy, inhomogeneous real world datasets, where weighting individual data points might be crucial for performance.

We report, to our knowledge, performance consistent with state-of-the-art in 1-shot and 5-shot classification both in 5-way and 20-way regime on the Omniglot dataset. (Lake et al., 2015) By studying the response of our model to partially down-sampled data, we hypothesize that its advantage might be more significant in lower quality, inhomogeneous datasets.

This paper is structured as follows: We describe related work in Section 2. We then proceed to introduce our methods in Section 3. The episodic training scheme is also presented there. We discuss the Omniglot dataset in Section 4, and our experiments in Section 5. Finally, our conclusions are presented in Section 6.

## 2    RELATED WORK

Non-parametric models, such as $k$-nearest neighbors (kNN), are ideal candidates for few-shot classifiers, as they allow for incorporation of previously unseen classes after training. However, they are very sensitive to the choice of distance metric. (Atkeson et al., 1997) Using the distance in the space of inputs directly (e.g. raw pixel values) does not produce high accuracies, as the connection between the image class and its pixels is very non-linear.

A straightforward modification in which a metric embedding is learned and then used for kNN classification has yielded good results, as demonstrated by Goldberger et al. (2005), Weinberger & Saul (2009), Kulis (2013), and Bellet et al. (2013). An approach using *matching networks* has been proposed in Vinyals et al. (2016), in effect learning a distance metric between pairs of images. A noteworthy feature of the method is its training scheme, where each mini-batch (called an *episode*) tries to mimic the data-poor test conditions by sub-sampling the number of classes as well as numbers of examples in each. It has been demonstrated that such an approach improves performance on few-shot classification. (Vinyals et al., 2016) We therefore use it as well.

Instead of learning on the dataset directly, it has recently been proposed (Ravi & Larochelle, 2017) to train an LSTM (Hochreiter & Schmidhuber, 1997) to predict updates to a few-shot classifier given an episode as its input. This approach is referred to as *meta-learning*. Meta-learning has been reaching high accuracies on Omniglot (Lake et al., 2015), as demonstrated by Finn et al. (2017), and Munkhdalai & Yu (2017). A task-agnostic meta-learner based on temporal convolutions has been proposed in Mishra et al. (2017), currently outperforming other approaches. Combinations of parametric and non-parametric methods have been the most successful in few-shot learning recently. (Snell et al., 2017)(Ren & Yu, 2014)(Koch et al., 2015)

Our approach is specific to classification of images, and does not attempt to solve the problem via meta-learning. We build on the model presented in Snell et al. (2017), which maps images into embedding vectors, and uses their clustering for classification. The novel feature of our model is that it predicts its confidence about individual data points via a learned, image-dependent covariance matrix. This allows it to construct a richer embedding space to which images are projected. Their clustering under a direction and class-dependent distance metric is then used for classification.

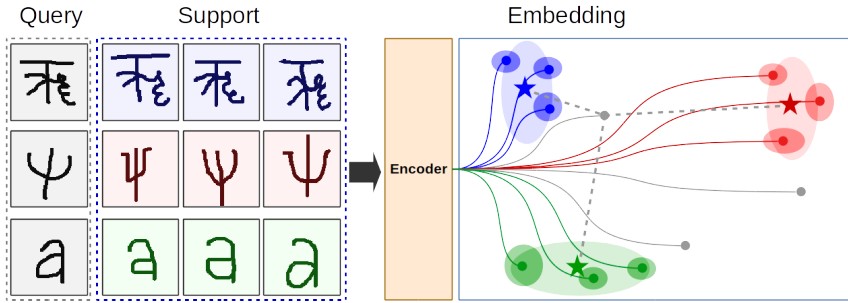

Figure 1: A diagram of the function of the Gaussian prototypical network. An encoder maps an image into a vector in the embedding space (dark circles). A covariance matrix is also output for each image (dark ellipses). Support images are used to define the prototypes (stars), and covariance matrices (light ellipses) of the particular class. The distances between centroids, and encoded query images, modified by the total covariance of a class, are used to classify query images. The distances are shown as dashed gray lines for a particular query point.

## 3 METHODS

In this paper, we build on *prototypical networks* described in Snell et al. (2017). We extend their architecture to what we call a *Gaussian prototypical network*, allowing the model to reflect quality of individual data points (images) by predicting their embedding vectors as well as confidence regions around them, characterized by a Gaussian covariance matrix.

A vanilla prototypical network comprises an encoder that maps an image into an embedding vector. A batch contains a subset of available training classes. In each iteration, images for each class are randomly split into *support*, and *query* images. The embeddings of support images are used to define class *prototypes* – embedding vectors typical for the class. The proximity of query image embeddings to the class prototypes is used for classification.

The encoder architectures of vanilla and Gaussian prototypical networks do not differ. The key difference is the way encoder outputs are interpreted, used, and how the metric on the embedding space is constructed. The function of the Gaussian network is presented in Figure 1.

### 3.1 ENCODER

We use a multi-layer convolutional neural network without an explicit, final fully connected layer to encode images into high-dimensional Euclidean vectors. For a vanilla prototypical network, the encoder is a function taking an image $I$ and transforming it into a vector $\vec{x}$ as

$$\text{encoder}(\theta) : I \in \mathbb{R}^{H \times W \times C} \to \vec{x} \in \mathbb{R}^D \,, \tag{1}$$

where $H$ and $W$ are the height and width of the input image, and $C$ is the number of its channels. $D$ is the embedding dimension of our vector space which is a hyperparameter of the model. $\theta$ are the trainable weights of the encoder.

For a Gaussian prototypical network, the output of the encoder is a concatenation of an embedding vector $\vec{x} \in \mathbb{R}^D$ and real vector $\vec{s}_{\text{raw}} \in \mathbb{R}^{D_S}$ relevant to the covariance matrix $\Sigma \in \mathbb{R}^{D \times D}$. Therefore

$$\text{encoder}_{\text{Gauss}}(\theta) : I \in \mathbb{R}^{H \times W \times C} \to [\vec{x}, \vec{s}_{\text{raw}}] \in \left[\mathbb{R}^D, \mathbb{R}^{D_S}\right] \,, \tag{2}$$

where $D_S$ is the number of degrees of freedom of the covariance matrix.

We explore three variants of the Gaussian prototypical network:

  a) **Radius** covariance estimate. $D_S = 1$ and only a single real number $s_{\text{raw}} \in \mathbb{R}^1$ is generated per image. As such the covariance matrix has the form $\Sigma = \text{diag}\left(\sigma, \sigma, \ldots, \sigma\right)$, where $\sigma$ is calculated from the raw encoder output $s_{\text{raw}}$. The confidence estimate is therefore not direction-sensitive.

b) **Diagonal** covariance estimate. $D_S = D$ and the dimension of the covariance estimate is the same as of the embedding space. $\vec{s}_{\text{raw}} \in \mathbb{R}^D$ is generated per image. Therefore the covariance matrix has the form $\Sigma = \text{diag}\,(\vec{\sigma})$, where $\vec{\sigma}$ is calculated from the raw encoder output $\vec{s}_{\text{raw}}$. This allows the network to express direction-dependent confidence about a data point.

c) **Full** covariance estimate. A full covariance matrix is output per data point. This method proved to be needlessly complex for the tasks given and therefore was not explored further.

We were using 2 encoder architectures: 1) a **small** architecture, and 2) a **big** architecture. The small architecture corresponded to the one used in Snell et al. (2017), and we used it to validate our own experiments with respect. The big architecture was used to see the effect of an increased model capacity on accuracy. As a basic building block, we used the sequence of layers in Equation 3.

$$3 \times 3\,\text{CNN} \to \text{batch normalization} \to \text{ReLU} \to 2 \times 2\,\text{max pool} \tag{3}$$

Both architectures were composed of 4 such blocks stacked together. The details of the architectures are as follows:

1) **Small architecture:** $3 \times 3$ filters, numbers of filters $[64, 64, 64, D]$ ($[64, 64, 64, D+1]$ for the radius Gaussian model, $[64, 64, 64, 2D]$ for the diagonal Gaussian model). Embedding space dimensions explored were $D = 32, 64, 128$.

2) **Big architecture:** $3 \times 3$ filters, numbers of filters $[128, 256, 512, D]$ ($[128, 256, 512, D+1]$ for the radius Gaussian model, $[128, 256, 512, 2D]$ for the diagonal Gaussian model). Embedding space dimensions explored were $D = 128, 256, 512$.

We explored 4 different methods of translating the raw covariance matrix output of the encoder into an actual covariance matrix. Since we primarily deal with the inverse of the covariance matrix $S = \Sigma^{-1}$, we were predicting it directly. Let the relevant part of the raw encoder output be $S_{\text{raw}}$. The methods are as follows:

a) $S = 1 + \text{softplus}\,(S_{\text{raw}})$, where $\text{softplus}(x) = \log\,(1 + e^x)$ and it is applied componentwise. Since $\text{softplus}(x) > 0$, this guarantees $S > 1$ and the encoder can only make data points less important. The value of $S$ is also not limited from above. Both of these approaches prove beneficial for training. Our best models used this regime for initial training.

b) $S = 1 + \text{sigmoid}\,(S_{\text{raw}})$, where $\text{sigmoid}(x) = 1/\,(1 + e^{-x})$ and it is applied componentwise. Since $\text{sigmoid}(x) > 0$, this guarantees $S > 1$ and the encoder can only make data points less important. The value of $S$ is bounded from above, as $S < 2$, and the encoder is therefore more constrained.

c) $S = 1 + 4\,\text{sigmoid}\,(S_{\text{raw}})$ and therefore $1 < S < 5$. We used this to explore the effect of the size of the domain of covariance estimates on performance.

d) $S = \text{offset} + \text{scale} \times \text{softplus}\,(S_{\text{raw}}/\text{div})$, where $\text{offset}$, $\text{scale}$, and $\text{div}$ are initialized to 1.0 and are trainable. Our best models used this regime for late-stage training, as it is more flexible and data-driven than a).

## 3.2 EPISODIC TRAINING

A key component of the prototypical model is the episodic training regime described in Snell et al. (2017) and modeled on Vinyals et al. (2016). During training, a subset of $N_c$ classes is chosen from the total number of classes in the training set (without replacement). For each of these classes, $N_s$ *support* examples are chosen at random, as well as $N_q$ *query* examples. The encoded embeddings of the support examples are used to define where a particular class *prototype* lies in the embedding space. The distances between the query examples and positions of class prototypes are used to classify the query examples and to calculate loss. For the Gaussian prototypical network, the covariance of each embedding point is estimated as well. A diagram of the process is shown in Figure 1.

For a Gaussian prototypical network, the radius or a diagonal of a covariance matrix is output together with the embedding vector (more precisely its raw form is, as detailed in Section 3.1). These are then used to weight the embedding vectors corresponding to support points of a particular class,

as well as to calculate a total covariance matrix for the class. The distance $d_c(i)$ from a class prototype $c$ to a query point $i$ is calculated as

$$d_c^2(i) = (\vec{x}_i - \vec{p}_c)^T S_c (\vec{x}_i - \vec{p}_c) , \tag{4}$$

where $\vec{p}_c$ is the centroid, or *prototype*, of the class $c$, and $S_c = \Sigma_c^{-1}$ is the inverse of its class covariance matrix. The Gaussian prototypical network is therefore able to learn class and direction-dependent distance metric in the embedding space. We found that the speed of training and its accuracy depend strongly on how distances are used to construct a loss. We conclude that the best option is to work with linear Euclidean distances, i.e. $d_c(i)$. The specific form of the loss function used is presented in Algorithm 1.

### 3.3 DEFINING A CLASS

A critical part of a prototypical network is the creation of a class prototype from the available support points of a particular class. We propose a variance-weighted linear combination of embedding vectors of individual support examples as our solution. Let class $c$ have support images $I_i$ that are encoded into embedding vectors $\vec{x}_i^c$, and inverses of covariance matrices $S_i^c$, whose diagonals are $\vec{s}_i^c$. The prototype is defined as

$$\vec{p}_c = \frac{\sum_i \vec{s}_i^c \circ \vec{x}_i^c}{\sum_i \vec{s}_i^c} , \tag{5}$$

where $\circ$ denotes a component-wise multiplication, and the division is also component-wise. The diagonal of the inverse of the class covariance matrix is then calculated as

$$\vec{s}_c = \sum_i \vec{s}_i^c . \tag{6}$$

This corresponds to the optimal combination of Gaussians centered on the individual points into an overall class Gaussian, hence the name of the network. The elements of $s$ are effectively $1/\sigma^2$. Equations 5 and 6 therefore correspond to weighting by $1/\sigma^2$. The full algorithm is described in Algorithm 1.

### 3.4 EVALUATING MODELS

To estimate the accuracy of a model on the test set, we classify the whole test set for every number of support points $N_s = k$ in the range $k \in [1, ..19]$. The number of query points for a particular $k$ is therefore $N_q = 20 - N_s$, as Omniglot provides 20 examples of each class. The accuracies are then aggregated, and for a particular stage of the model training a $k$-shot classification accuracy as a function of $k$ is determined. Since we are not using a designated validation set, we ensure our impartiality by considering the test results for the 5 highest training accuracies, and calculate their mean and standard deviation. By doing that, we prevent optimizing our result for the test set, and furthermore obtain error bounds on the resulting accuracies. We evaluate our models in 5-way and 20-way test classification to directly compare to existing literature.

## 4 DATASETS

We used the Omniglot dataset. (Lake et al., 2015) Omniglot contains 1623 character classes from 50 alphabets (real and fictional) and 20 hand-written, gray-scale, $105 \times 105$ pixel examples of each. We down-sampled them to $28 \times 28 \times 1$, subtracted their mean, and inverted them. We were using the recommended split to 30 training alphabets, and 20 test alphabets, as suggested by Lake et al. (2015), and used by Snell et al. (2017). The training set included overall 964 unique character classes, and the test set 659 of them. There was no class overlap between the training and test datasets. We did not use a separate validation set as we did not fine-tune hyperparameters and chose the best performing model based on training accuracies alone (see Section 3.4).

To extend the number of classes, we augmented the dataset by rotating each character by $90°$, $180°$, and $270°$, and defined each rotation to be a new character class on its own. The same approach is used in Vinyals et al. (2016), and Snell et al. (2017). An example of an augmented character is shown in Figure 2. This increased the number of classes 4-fold. In total, there were 77,120 images

---

**Algorithm 1** Classification and loss algorithm for Gaussian prototypical networks

---

**Input:** Images $I$, class labels $y$, encoder $f$, $N_s$ number of support points per class, $N_q$ number of query points per class, $N_c$ number of classes in a batch
**Output:** Predicted labels $\hat{y}$, loss $L$
  **for** batch in data **do**
      Choose a subset $C_{\text{used}}$ of $N_c$ classes from all possible training classes at random (without replacement)
      **for** class $c$ in classes $C_{\text{used}}$ **do**
         Choose $N_s$ *support* examples for the class $c$ from batch and call them $S_c$.
         Choose $N_q$ *query* examples for the class $c$ from batch and call them $Q_c$.
         Embed $Q_c$ as $f(Q_c) \to \tilde{Q}^c,\ _-$
         Embed $S_c$ as $f(S_c) \to \tilde{S}^c, \{\vec{s}^{\mathrm{c}}\}$              ▷ vectors and covariances
         Class prototype $\vec{p}_c \leftarrow \left(\sum_i \vec{s}_i^{\mathrm{c}} \circ \tilde{S}_i^c\right) / \left(\sum_i \vec{s}_i^{\mathrm{c}}\right)$    ▷ summed over support points
         Class inverse covariance $\vec{s}^{\mathrm{c}} = \sum_i \vec{s}_i^{\mathrm{c}}$         ▷ summed over support points
      **end for**
      Loss $L \leftarrow 0$              ▷ zeroing the loss per batch
      **for** query image $i$ in batch query points **do**
         Let the image $i$ embedding vector be $\vec{x}$ and its true class $y$
         **for** class $c$ in classes $C_{\text{used}}$ **do**
            difference$(c, i) = \vec{x} - \vec{p}_c$        ▷ position difference from class $c$
            distance$(c, i) = \sqrt{(\vec{x} - \vec{p}_c)^T \vec{s}^{\mathrm{c}} \circ (\vec{x} - \vec{p}_c)}$   ▷ covariance-modified distance
         **end for**
         Predicted label $\hat{y}_i \leftarrow \arg\min_c \text{distance}(c, i)$   ▷ labelled based on the "closest" prototype
         $L \leftarrow L + \frac{1}{N_c}\text{softmax cross entropy}(-\text{distance}(c, i), y)$   ▷ summed over $c$
      **end for**
      $L \leftarrow L/\text{batch size}$          ▷ loss per batch for comparability
  **end for**

---

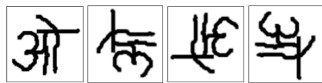

Figure 2: An example of augmentation of class count by rotations. An original character (on the left) is rotated by 90°, 180°, and 270°. Each rotation is then defined as a new class. This enhances the number of classes, but also introduces degeneracies for symmetric characters.

in the training set, and 52,720 images in the test set. Due to the rotational augmentation, characters that have a rotational symmetry were nonetheless defined as multiple classes. As even a hypothetical perfect classifier would not be able to differentiate e.g. the character "O" from a rotated "O", 100 % accuracy was not reachable.

## 5 EXPERIMENTS

We conducted a large number of few-shot learning experiments on the Omniglot dataset. For Gaussian prototypical networks, we explored different embedding space dimensionalities, ways of generating the covariance matrix, and encoder capacities (see Section 3.1 for details). We also compared them to vanilla prototypical networks, and showed that our Gaussian variant is a favorable way of using additional trainable parameters compared to increasing embedding space dimensionality. We found that predicting a single number per embedding point (the radius method in Section 3.1) works the best on Omniglot.

In general, we explored the size of the encoder (small, and big, as described in Section 3), the Gaussian/vanilla prototypical network comparison, the distance metric (cosine, $\sqrt{L_2}$, $L_2$, and $L_2^2$), the number of degrees of freedom of the covariance matrix in the Gaussian networks (radius, and diagonal estimates, see Section 3.1), and the dimensionality of the embedding space. We also explored

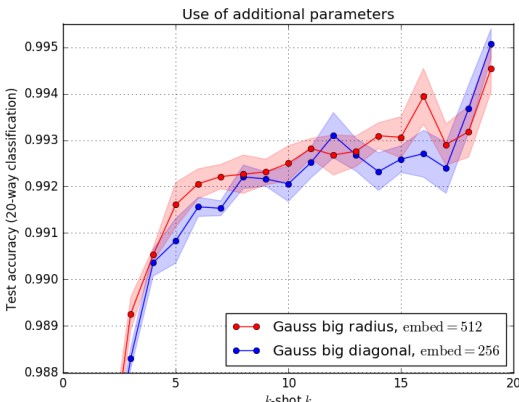

Figure 3: Comparison of two methods of allocation of extra parameters. Allocating extra parameters to increase embedding space dimensionality (radius), or making a more precise covariance estimate (diagonal). The radius estimate (1 additional real number per embedding vector) outperforms the diagonal estimate with the same number of parameters.

augmenting the input dataset by down-sampling a subset of it to encourage the usage of covariance estimates by the network, and found that this improves $(k > 1)$-shot performance.

We were using the *Adam* optimizer with an initial learning rate of $2 \times 10^{-3}$. We halved the learning rate every 2000 episodes $\approx 30$ epochs. All our models were implemented in `TensorFlow`, and ran on a single NVidia K80 GPU in Google Cloud. The training time of each model was less than a day.

We trained our models with $N_c = 60$ classes (60-way classification) at training time, and tested on $N_{ct} = 20$ classes (20-way) classification. For our best-performing models, we also conducted a final $N_{ct} = 5$ (5-way) classification test to compare our results to literature. During training, each class present in the mini-batch comprised $N_s = 1$ support points, as we found that limiting the number of support points leads to better accuracies. This could intuitively be understood as matching the training regime to the test regime, as done in Vinyals et al. (2016). The remaining $N_q = 20 - N_s = 19$ images per class were used as query points.

We verified, provided that the covariance estimate is not needlessly complex, that using encoder outputs as covariance estimates is more advantageous than using the same number of parameters as additional embedding dimension. This holds true for the *radius* estimate (i.e. one real number per embedding vector), however, the *diagonal* estimate does not seem to help with performance (keeping the number of parameters equal). This effect is shown in Figure 3.

The best performing model was initially trained on the undamaged dataset for 220 epochs. The training then continued with 1.5% of images down-sampled to $24 \times 24$, 1.0% down-sampled to $20 \times 20$, and 0.5% down-sampled to $16 \times 16$ for 100 epochs. Then with 1.5% down-sampled to $23 \times 23$ and 1.0% down-sampled to $17 \times 17$ for 20 epochs, and 1.0% down-sampled to $23 \times 23$ for 10 epochs. These choices were quite arbitrary and not optimized over. The purposeful damage to the dataset encouraged usage of the covariance estimate and increased $(k > 1)$-shot results, as demonstrated in Figure 4.

The comparison of our models to results from literature is presented in Table 1. To our knowledge, our models perform consistently with state-of-the-art in 1-shot and 5-shot classification both in 5-way and 20-way regime on the Omniglot dataset. In 5-shot 5-way classification in particular, we are reaching very close to perfect performance ($99.73 \pm 0.04$ %) and therefore conclude that a more complex dataset is needed for further few-shot learning algorithms development.

## 5.1 Usage of covariance estimate

In order to validate our assumption that the Gaussian prototypical network outperforms the vanilla version due to its ability to predict covariances of individual embedded images and therefore the

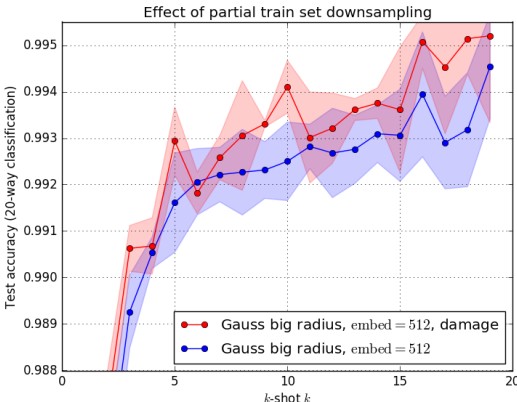

Figure 4: The effect of down-sampling a part of the training set on $k$-shot test accuracy. The network trained on purposefully damaged data outperforms the one trained on unmodified data, as it learns to utilize covariance estimates better.

Table 1: The results of our experiments as compared to other papers. To our knowledge, our models perform consistently with state-of-the-art in 1-shot and 5-shot classification both in 5-way and 20-way regime on the Omniglot dataset.

| Model | 20-way | | 5-way | |
|---|---|---|---|---|
| | 1-shot | 5-shot | 1-shot | 5-shot |
| Matching networks (Vinyals et al., 2016) | 93.8% | 98.5% | 98.1% | 98.9% |
| Matching networks (Vinyals et al., 2016) | 93.5% | 98.7% | 97.9% | 98.7% |
| Neural statistician (Edwards & Storkey, 2016) | 93.2% | 98.1% | 98.1% | 99.5% |
| Prototypical network (Snell et al., 2017) | 96.0% | 98.9% | 98.8% | **99.7%** |
| Finn et al. (2017) | | | 98.7 ± 0.4% | 99.9 ± 0.3% |
| Munkhdalai & Yu (2017) | | | 98.9% | |
| TCML (Mishra et al., 2017) | **97.64 ± 0.30%** | **99.36 ± 0.18%** | 98.96 ± 0.20% | **99.75 ± 0.11%** |
| Gauss (radius) (ours) | 97.02 ± 0.40% | **99.16 ± 0.11%** | **99.02 ± 0.11%** | 99.66 ± 0.04% |
| Gauss (radius) damage (ours) | 96.94 ± 0.31% | **99.29 ± 0.09%** | **99.07 ± 0.07%** | **99.73 ± 0.04%** |

possibility to down-weight them, we studied the distribution of predicted values of $s$ (see Section 3.1 for details) for our best performing network for undamaged and damaged test data. The network was trained on partially down-sampled training data.

For the undamaged test set, the vast majority of covariance estimates took the same value, indicating that the network did not use its ability to down-weight data points. However, for a partially down-sampled test set, the distribution of magnitudes of covariance estimates got significantly broader. We interpret this as a confirmation that the network learned to put less emphasis on down-sampled images. A comparison of both distributions is shown in Figure 5.

## 6  CONCLUSION

In this paper we proposed Gaussian prototypical networks for few-shot classification – an improved architecture based on prototypical networks (Snell et al., 2017). We tested our models on the Omniglot dataset, and explored different approaches to generating a covariance matrix estimate together with an embedding vector. We showed that Gaussian prototypical networks outperform vanilla prototypical networks with a comparable number of parameters, and therefore that our architecture choice is beneficial. We found that estimating a single real number on top of an embedding vector works better than estimating a diagonal, or a full covariance matrix. We suspect that lower quality, less homogeneous datasets might prefer a more complex covariance matrix estimate.

Contrary to Snell et al. (2017), we found that the best results are obtained by training in the 1-shot regime. Our results are consistent with state-of-the-art in 1-shot and 5-shot classification both in

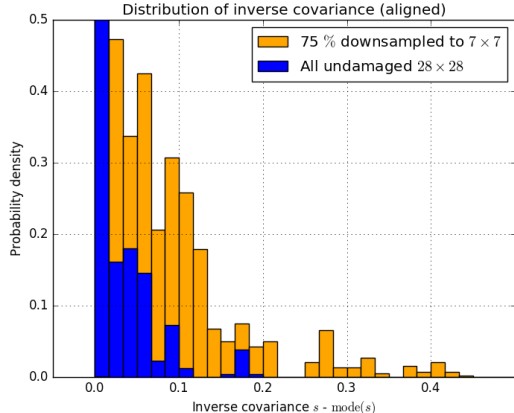

Figure 5: Predicted covariances for the original test set and a partially down-sampled version of it. The Gaussian network learned to down-weight damaged examples by predicting a higher $s$, as apparent from the heavier tail of the yellow distribution. The distributions are aligned together, as only the difference between the leading edge and a value influence classification.

5-way and 20-way regime on the Omniglot dataset. Especially for 5-way classification, our results are very close to perfect performance.

We got better accuracies (in particular for $(k > 1)$-shot classification) by artificially down-sampling fractions of our training dataset, encouraging the network to fully utilize covariance estimates. We hypothesize that the ability to learn the embedding as well as its uncertainty would be even more beneficial for poorer-quality, heterogeneous datasets, which are commonplace in real world applications. There, down-weighting some data points might be crucial for faithful classification. This is supported by our experiments with down-sampling Omniglot.

ACKNOWLEDGMENTS

We would like to thank Ben Poole, Yihui Quek, and Justin Johnson (all at Stanford University) for useful discussions. A part of this work was done as a class project for the Stanford University CS 231N: *Convolutional Neural Networks for Visual Recognition*, which provided Google Cloud credit coupons that partially supported our GPU usage.

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
