# OpenReview forum: "Gaussian Prototypical Networks for Few-Shot Learning on Omniglot"
_ICLR.cc/2018/Conference — Reject_

### Official Review · AnonReviewer2 · 2017-11-26
**An interesting direction, but the presentation is unclear at a few places, and need more experiments/analysis.**

**Rating:** 4
**Confidence:** 4

**Review:**

This paper presents an interesting extension to Snell et al.'s prototypical networks, by introducing uncertainty through a parameterised estimation of covariance along side the image embeddings (means). Uncertainty may be particularly important in the few-shot learning case this paper examines, when it is helpful to extract more information from limited number of input samples.

However, several important concepts in the paper are not well explained or motivated. For example, it is a bit misleading to use the word "covariance" throughout the paper, when the best model only employs a scalar estimate of the variance. A related, and potentially technical problem is in computing the prototype's mean and variance (section 3.3). Eq. 5 and 6 are not well motivated, and the claim of "optimal" under eq.6 is not explained. More importantly, eq. 5 and 6 do not use any covariance information (off-diagonal elements of S) --- as a result, the model is likely to ignore the covariance structure even when using full covariance estimate. The distance function (eq. 4) is d Mahalanobis distance, instead of "linear Euclidean distance". While the paper emphasises the importance of the form of loss function, the loss function used in the model is given without explanation (and using cross-entropy over distances looks hacky).

In addition, the experiments are too limited to support the claimed benefits from encoding uncertainty. Since the accuracies on omniglot data from recent models are already close to perfect, it is unclear whether the marginally improved number reported here is significant. In addition, more analysis may better support existing claims. For example, showing subsampled images indeed had higher uncertainty, rather than only the histogram for all data points.

Pros:
-Interesting problem and interesting direction.
-Considers a number of possible alternative models
-Intuitive illustration in Fig. 1

Cons:
-Misleading use of "covariance"
-The several important concepts including prototype mean/variance, distance, and loss are not well motivated or explained
-Evaluation is too limited

---

### Official Review · AnonReviewer1 · 2017-11-26
**Extension of prototypical networks of Snell et al. NIPS 2017 to learn a metric matrix per instance in addition to the instance projection.**

**Rating:** 3
**Confidence:** 4

**Review:**

The paper extends the prototypical networks of Snell et al, NIPS 2017 for one shot learning. Snell et al use a soft kNN classification rule, typically used in standard metric learning work (e.g. NCA, MCML), over learned instance projections, i.e. distances are computed over the learned projections. Each class is represented by a class prototype which is given by the average of the projections of the class instances. Classification is done with soft k-NN on the class prototypes. The distance that is used is the Euclidean distance over the learned representations, i.e. (z-c)^T(z-c), where z is the projection of the x instance to be classified and c is a class prototype, computed as the average of the projections of the support instances of a given class.

The present paper extends the above work to include the learning of a Mahalanobis matrix, S, for each instance, in addition to learning its projection. Thus now the classification is based on the Mahalanobis distance: (z-c)^T S_c (z-c). On a conceptual level since S_c should be a PSD matrix it can be written as the square of some matrix, i.e. S_c = A_c^TA_c, then the Mahanalobis distance becomes (A_c z - A_c c)^T ( A_c z-A_c c), i.e. in addition to learning a projection as it is done in Snell et al, the authors now learn also a linear transformation matrix which is a function of the support points (i.e. the ones which give rise to the class prototypes). The interesting part here is that the linear projection is a function of the support points. I wonder though if such a transformation could not be learned by the vanilla prototypical networks simply by learning now a projection matrix A_z as a function of the query point z. I am not sure I see any reason why the vanilla prototypical networks cannot learn to project x directly to A_z z and why one would need to do this indirectly through the use of the Mahalanobis distance as proposed in this paper.

On a more technical level the properties of the learned Mahalanobis matrix, i.e. the fact that it should be PSD, are not really discussed neither how this can be enforced especially in the case where S is a full matrix (even though the authors state that this method was not further explored). If S is diagonal then the S generation methods a) b) c) in the end of section 3.1 will make sure that S is PSD, I do not think that this is the case with d) though.

In the definition of the prototypes the component wise weigthing (eq. 5) works when the Mahalanobis matrix is diagonal (even though the weighting should be done by the \sqrt of it), how would it work if it was a full matrix is not clear.

On the experiments side the authors could have also experimented with miniImageNet and not only omniglot as is the standard practice in one shot learning papers.

I am not sure I understand figure 3 in which the authors try to see what happens if instead of learning the Mahalanobis matrix one would learn a projection that would have as many additional dimensions as free elements in the Mahalanobis matrix. I would expect to see a comparison of the vanilla prototypical nets against their method for each one of the different scenarios of the free parameters of the S matrix, something like a ratio of accuracies of the two methods in order to establish whether learning the Mahalanobis matrix brings an improvement over the prototypical nets with an equal number of output parameters.

---

### Official Review · AnonReviewer3 · 2017-11-27
**A potentially interesting idea for modelling uncertainty in prototype learning is proposed. However, the novelty of the the method is unclear,and the motivation is a bit confusing. Unfortunately, the experiments contain only one specific datasets, and the significance of the results is difficult to interpret.**

**Rating:** 3
**Confidence:** 4

**Review:**

SUMMARY: This work is about prototype networks for image classification. The idea is to jointly embed an image and a "confidence measure" into a latent space, and to use these embeddings to define prototypes together with confidence estimates. A Gaussian model is used for representing these confidences as covariance matrices. Within a class, the inverse covariance matrices of all corresponding images are averaged to for the inverse class-specific matrix S-C, and this S_C defines the tensor in the Mahalanobis metric for measuring the distances to the prototype.

EVALUATION:
CLARITY: I found the paper difficult to read. In principle, the idea seems to be clear, but then the description and motivation of the model remains very vague. For instance, what is the the precise meaning of an image-specific covariance matrix (supported by just one point)? What is the motivation to just average the inverse covariance matrices to compute S_C? Why isn't the covariance matrix estimated in the usual way as the empirical covariance in the embedding space?
NOVELTY: Honestly, I had difficulties to see which parts of this work could be sufficiently novel. The idea of using a Gaussian model and its associated Mahalanobis metric is certainly interesting, but also a time-honored concept. The experiments focus very specifically on the omniglot dataset, and it is not entirely clear to me what  should be concluded from the results presented. Are you sure that there is any significant improvement over the models in (Snell et al, Mishra et al, Munkhandalai & Yu, Finn et al.)?

---

### Decision · Program_Chairs · 2018-01-29
**ICLR 2018 Conference Acceptance Decision**

**Decision:**

Reject

**Comment:**

The reviewers agree that the idea of utilizing covariance information in the few-shot setting is interesting. There are concerns with the novelty of the paper, as well as the correctness in terms of ensuring the covariance matrix is PSD in all cases. There are some concerns with the experimental evaluation as well. In this area, Omniglot is a good sanity check, but other baseline datasets like miniImagenet are necessary to determine if this approach is truly useful.